# A single-domain green fluorescent protein catenane

Zhiyu Qu[1,2,3,4], Jing Fang[1,2,3,4], Yu-Xiang Wang[1,2,3,4], Yibin Sun[1,2,3,4], Yajie Liu[1,2,3,4], Wen-Hao Wu[1,2,3,4] & Wen-Bin Zhang [1,2,3,4,5] ✉

Natural proteins exhibit rich structural diversity based on the folds of an invariably linear chain. Macromolecular catenanes that cooperatively fold into a single domain do not belong to the current protein universe, and their design and synthesis open new territories in chemistry. Here, we report the design, synthesis, and properties of a single-domain green fluorescent protein catenane via rewiring the connectivity of GFP's secondary motifs. The synthesis could be achieved in two steps via a pseudorotaxane intermediate or directly via expression in cellulo. Various proteins-of-interest may be inserted at the loop regions to give fusion protein catenanes where the two subunits exhibit enhanced thermal resilience, thermal stability, and mechanical stability due to strong conformational coupling. The strategy can be applied to other proteins with similar fold, giving rise to a family of single-domain fluorescent proteins. The results imply that there may be multiple protein topological variants with desirable functional traits beyond their corresponding linear protein counterparts, which are now made accessible and fully open for exploration.

In Nature, nascent protein chains are endowed with a linear topology by ribosomal translation, upon which the current protein universe is built. With only a small set of native folds, natural proteins demonstrate considerable structural and functional diversity. However, macromolecular topology is by no means bound to be linear. Originally conceived in mathematics, topology has found prominent roles in a broad range of topics in chemistry, physics, and biology[1–5]. As a unique molecular attribute, it mainly concerns the connectivity and spatial relationship of a molecule[6,7]. Four basic macromolecular architectures, such as branched structures[8], (multi)cyclic structures[9], knots[3,10], and links[11–14], can be identified accordingly. Out of the vast majority of natural peptides or proteins, only a small fraction of them gain nontrivial topologies through chain folding and/or post-translational processing[15,16], which is thought to bring in certain functional benefits, such as stabilization[17–19].

In recent years, efforts have been made to construct artificial topological proteins, such as lasso proteins[20], protein pseudorotaxanes[21], and protein catenanes[22–26]. Their syntheses require a synergy between assembly motifs (which controls spatial relationships, such as p53 homodimer[22–24,27] and its heterodimer variant[25]) and reaction motifs[28] (which defines site-specific connectivity, such as SpyTag/SpyCatcher ligation[26] and split-intein mediated ligation[22]). Since both motifs are often foreign to the proteins-of-interest (POI), the resulting topological proteins are usually multi-domain and contain relics from entangling templates and/or scars from chemical ligation (Supplementary Fig. 1). Since most proteins are free of entanglement, it represents a grand challenge to design and synthesize a protein catenane that can cooperatively fold into a single domain.

Herein, we report a versatile strategy for the design and synthesis of single-domain protein catenanes, using green fluorescent protein (GFP) as a model protein, by rewiring the connectivity between different secondary motifs to introduce endogenous entanglement within a single domain without disturbing the overall folded structure (Fig. 1a). Following this tactic, a family of single-domain fluorescent protein (FP) catenanes (cat-FP) was constructed. It also offers a scaffold for making fusion protein catenanes (cat-GFP-POI) (Fig. 1b). These

[1]Beijing National Laboratory for Molecular Sciences, Beijing, P. R. China. [2]Key Laboratory of Polymer Chemistry & Physics of Ministry of Education, Peking University, Beijing, P. R. China. [3]Center for Soft Matter Science and Engineering, Peking University, Beijing, P. R. China. [4]College of Chemistry and Molecular Engineering, Peking University, Beijing, P. R. China. [5]Beijing Academy of Artificial Intelligence, Beijing, P. R. China. ✉e-mail: wenbin@pku.edu.cn

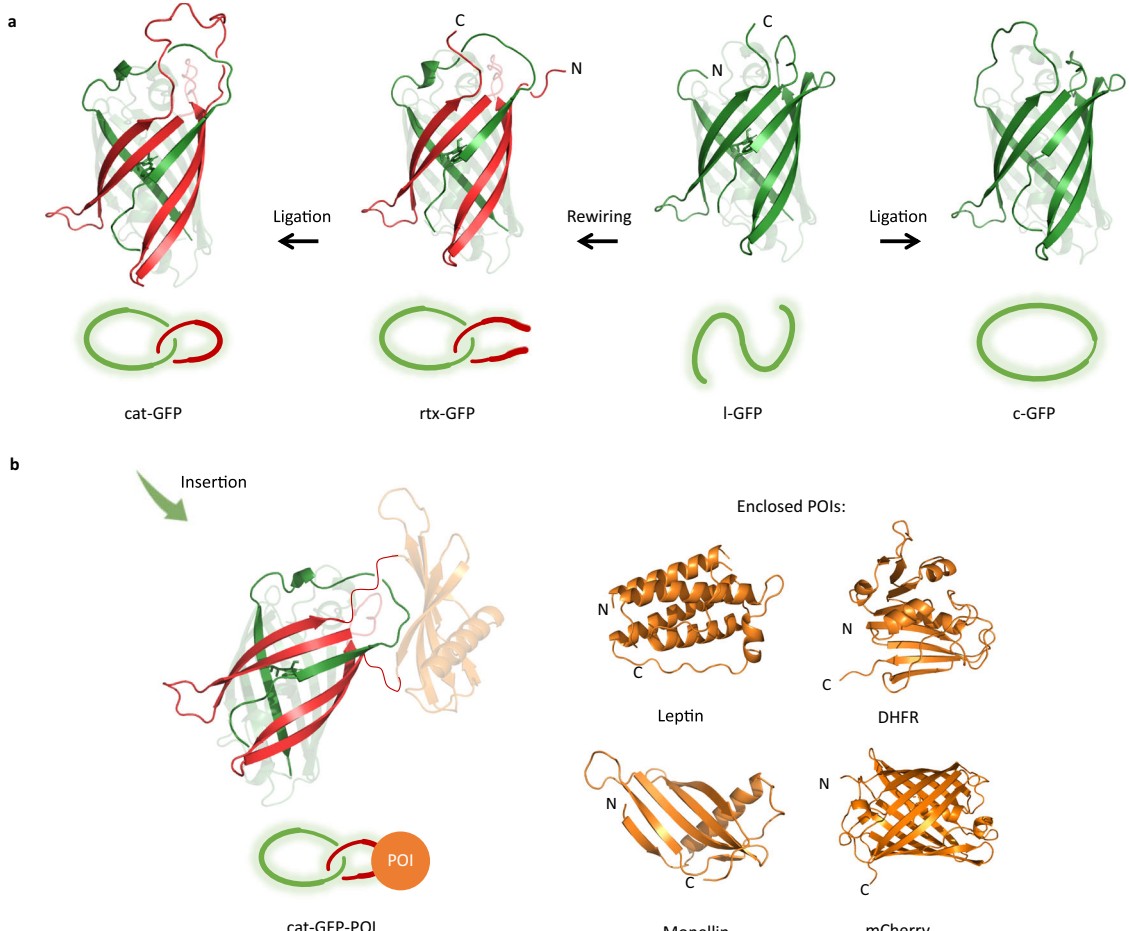

**Fig. 1 | Overview for topology engineering of green fluorescent protein.**
**a** Construction of single-domain protein catenane (cat-GFP), pseudorotaxane (rtx-GFP), and cyclic control (c-GFP) through rewiring and ligating wt-GFP (l-GFP). Two components constituting cat-GFP or rtx-GFP are colored red and green, respectively. N and C represents the N-terminal and C-terminal, respectively. **b** cat-GFP as a scaffold for fusion protein catenanes. Two components from GFP are colored red and green, protein of interest (POI) is colored orange. N and C represent the N-terminal and C-terminal, respectively.

samples allow us to further probe the effect of topology on structure-property relationships by comparison to their linear and cyclic controls.

## Results

### Design of single-domain GFP catenanes

A viable rewiring strategy toward a single-domain protein catenane requires not only the right ligation at specific sites (defining connectivity) but also the right entanglement between secondary elements (defining spatial relationship). The following rules-of-thumb were followed: (1) the split and rewire sites should be as few as possible and carefully chosen to ensure high reconstitution efficiency; (2) the new linkers should closely resemble the native ones to help define specific spatial relationship of crossing by their length and sequence; (3) synthetic methods should be deliberately designed with synergy between fold and reaction to enable selective synthesis of a specific topology. The superfolder-OPT variant[29] of GFP was chosen as a model protein because it folds fast and robustly into a conserved β-barrel structure even after extensive engineering[30] and the correct folding and activity can be easily assayed by fluorescence[31,32].

Theoretically, if the 11 β-strands of GFP are kept intact, there are 1,062,864 (~10⁶) ways to reconnect them into two individual rings as components of a catenane. In reality, it is unnecessary to exhaust all these possibilities since the reconstitution efficiency drops rapidly with increasing extent of engineering. If only one loop is broken for rewiring, there are just 10 different combinations. The number rises to

135 ($C^2_{10}C^1_3$) when two loops are engineered. By visual inspection of the GFP structure (PDB: 2B3P) by PyMOL, we came up with 10 possible catenane designs (Supplementary Table 1, see also Supplementary Fig. 2a–j for their topology diagrams). Design A seems to have the highest probability of success, as only one loop is split (at 156 K/157Q) and rewired to form two interlocked rings (strand 1–7 and 8–11) with a size ratio of ~2:1 (Fig. 2). The structure of Design A predicted by AlphaFold 2 (AF2)[33] indeed shows entanglement between two rings, which urged us to further pursue their synthesis.

### Two-step synthesis of cat-GFP

Assembly-reaction synergy has been demonstrated as an effective strategy to prepare nonlinear proteins via post-translational processing. We envision that the synthesis would involve first reconstitution between two fragments to form an entwined complex followed by ring closure to give the catenane, either stepwise via a preformed pseudorotaxane complex[26] or in one-pot using orthogonal ligation chemistry[22]. Although the latter appears to be simple in practice, we chose to begin with the former because the intermediate could be separated and well characterized and it is also possible to obtain higher order protein [n]catenanes from intermolecular reactions. Therefore, Design A without circular permutation was synthesized via a pseudorotaxane intermediate (Fig. 3a).

Briefly, the gene cassette was designed to be *IntC-GFP1-IntN-TVMV-GG-GFP2-LPETGGH₆*, or *IntC-GFP1-IntN-GFP2* for short, where *GFP1* stands for gene encoding strands 1–7 constituting the first ring,

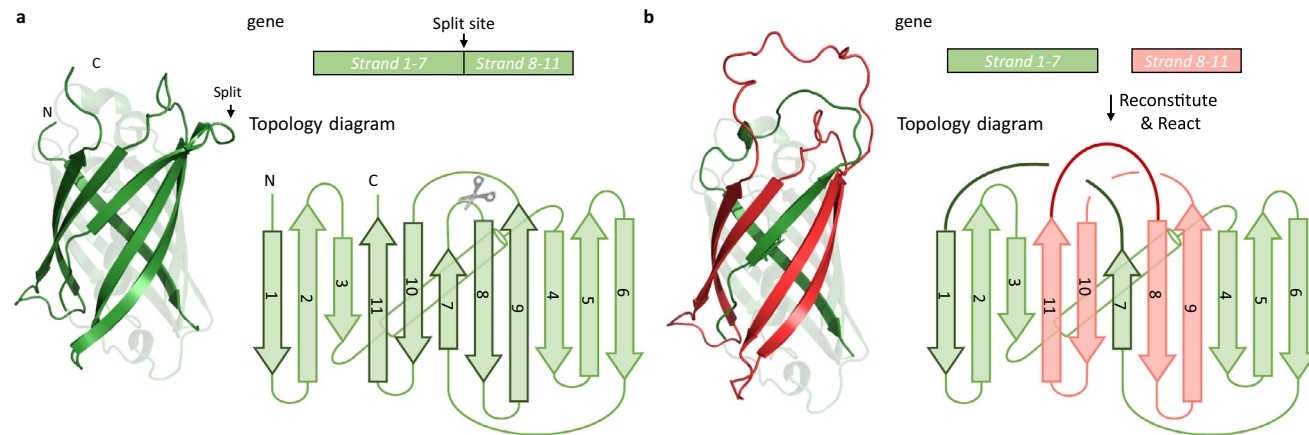

**Fig. 2 | Design of a single-domain GFP catenane. a** Cartoon illustration, topology diagram, and gene encoding the original GFP (PDB: 2B3P). Strands are numbered from N-terminal to C-terminal. N and C represent the N-terminal and C-terminal, respectively. **b** Cartoon illustration, topology diagram, and gene encoding single-domain GFP catenane. The solid bold lines in dark red and dark green represent the newly introduced linkers. Two rings constituting protein catenane are colored in red and green, respectively.

*GFP2* for gene encoding strands 8–11 constituting the second ring, *TVMV* for gene encoding proteolytic site (ETVRFQ/G) of tobacco vein mottling virus protease, *GGG* and *LPETG* for gene encoding the sortase recognition sites, *IntC* and *IntN* for gene encoding the split-intein pair, and *H₆* for gene encoding the His-tag (Supplementary sequence 1). The TVMV protease gene was constitutively co-expressed with this gene cassette using the pMCSG19/pRK1037 system[34]. Upon expression, the linear chain was converted to two fragments by TVMV, exposing N-terminal polyglycine tag for subsequent sortase-mediated ligation. Only GFP1 was cyclized in cellulo and GFP2 would be cyclized later so as to position the newly formed linker of GFP2 well above the linker in GFP1, forming the correct spatial relationship. Hence, the reconstitution of GFP1 and GFP2 and the cyclization of GFP1 by split intein pair would give a pseudorotaxane complex (rtx-GFP). The expression conditions were set to be 16 °C for 12 h to ensure GFP chromophore maturation and complete TVMV digestion. The culture turned greenish after 3–4 h upon induction. After purification first by Ni-NTA affinity chromatography and size exclusion chromatography (SEC), rtx-GFP was obtained as a stable complex with similar retention volume to wt-GFP (Supplementary Fig. 3). Both SDS-PAGE and LC-MS clearly show that rtx-GFP is composed of two components with molecular weights of ~11 kDa (l-GFP2) and ~20 kDa (c-GFP1) (Fig. 3b and Supplementary Fig. 3b). Notably, the apparently higher mobility of cyclic GFP1 in SDS-PAGE with a band at ~17 kDa is probably due to the macrocycle's compact conformation[35].

After that, rtx-GFP was subjected to sortase A mediated ligation in vitro. Sortase A recognizes the LPXTG motif (X = any canonical amino acid) and the N-terminal polyglycine tag and catalyzes the transpeptidation for backbone ligation[36,37]. As shown in Fig. 3b, rtx-GFP was converted to [2]catenane (denoted as [2]cat-GFP) through intramolecular reaction and [3]catenane (denoted as [3]cat-GFP) through intermolecular reaction, yielding 83% protein catenanes (46% for [2]cat-GFP and 37% for [3]cat-GFP) after Ni-NTA purification. [2]cat-GFP and [3]cat-GFP were separated using SEC (Fig. 3c) and their molecular weights were confirmed by LC-MS spectra (Fig. 3h). To prove the topology, a TEV digestion site (ENLYFQ/G) was placed on the GFP1 ring. Upon digestion by TEV protease, [2]cat-GFP and [3]cat-GFP were converted to a linearized GFP1 (l-GFP1) of ~20 kDa and a cyclic product at ~10 kDa (c-GFP2) for [2]cat-GFP and ~20 kDa (c-GFP2-GFP2) for [3]cat-GFP, respectively, as evidenced both by SDS-PAGE (Fig. 3d, e) and LC-MS (Supplementary Fig. 4), thus proving the catenane topology.

Both catenanes show identical fluorescent emission and excitation profiles to that of wt-GFP, suggesting similarly folded structures

with mature chromophores (Supplementary Fig. 5). This is consistent with the predicted structure of [2]cat-GFP, showing a very similar overall folding structure with wt-GFP (Fig. 3f). The structure of [3]cat-GFP was also predicted by AF2. The calculated overall structure (Fig. 3g) reveals two GFP1 rings (colored green) mechanically interlocked with a ring of two consecutive GFP2 fragments (colored red) spanning two GFP domains. Through this two-step route, we obtained a two-ring, single-domain GFP and a three-ring, two-domain GFP dimer with minimal foreign components.

### A family of single-domain cat-FPs

The fluorescent protein family contains diverse FP homologs from different organisms with a conserved β-barrel fold[38]. We hypothesized that the above strategy could be applied to other members of the FP family to give a family of single-domain cat-FPs. Specifically, mCherry originated from *Discosoma* sp.[39], mWasabi originated from *Clavularia* sp.[40], and YPet originated from *Aequorea Victoria*[41] were selected and similarly engineered to give the corresponding single-domain catenanes following the two-step synthesis method (Fig. 3a and Supplementary Fig. 6). Both LC-MS spectra and TEV digestion experiments have demonstrated the identity and topology of these cat-FPs (Fig. 3i and Supplementary Fig. 7). Notably, their excitation and emission profiles remain identical to their linear counterparts, indicating well-folded structures and fully matured chromophores (Supplementary Fig. 8). The results suggest the general applicability of this strategy for proteins with similar folds, which further implies that the linear FP family can be mapped onto the catenane FP family in a parallel protein universe.

### A robust scaffold for fusion protein catenanes

As shown in the structure of rtx-GFP predicted by AF2, the dangling chain ends for ring closure are located on the same side of the β-barrel with a distance of ~43 Å (Fig. 4a). Hence, we envision that it might well tolerate the insertion of a protein-of-interest (POI) and could serve as a robust scaffold for fusion protein catenanes. Four proteins with different folds, namely dihydrofolate reductase (DHFR)[42], leptin[43], monellin[44], and mCherry[45] were chosen as POIs. Their genes were cloned after GFP2 to give *IntC-GFP1-IntN-GFP2-POI*, respectively (Fig. 4a and Supplementary sequence 2). Cellular expression afforded the pseudorotaxane intermediates (rtx-GFP-POIs), which were further converted to fusion protein catenanes (cat-GFP-POIs) and thoroughly characterized (Fig. 4 and Supplementary Fig. 9). Notably, the selectivity for [2]catenanes was dramatically improved owing to the steric

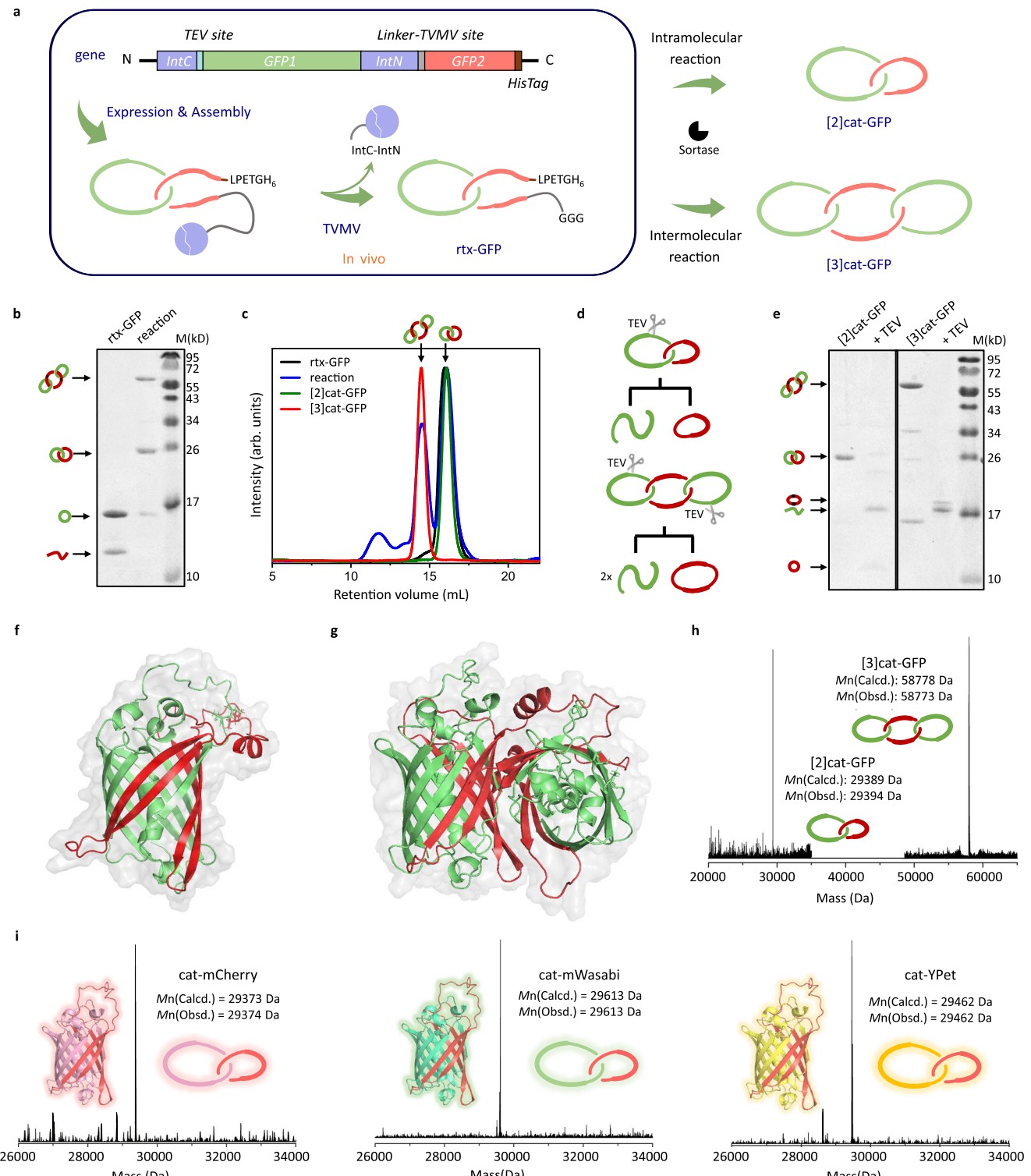

**Fig. 3 | Synthesis and characterization of FP catenanes. a** Two-step synthesis of GFP catenanes via a pseudorotaxane intermediate. Two rings constituting cat-GFP are colored red and green, respectively. **b** SDS-PAGE analysis of rtx-GFP (left lane) and the reaction mixture (right lane), showing the yields of in vitro synthesis (~46% for [2]cat-GFP and ~37% for [3]cat-GFP as determined by gel densitometry). **c** SEC overlay of rtx-GFP (black), the reaction mixture (blue), purified [2]cat-GFP (green) and [3]cat-GFP (red). **d** Cartoon illustration of corresponding TEV protease digestion products. **e** SDS-PAGE analysis of the TEV protease digestion products.

**f** Structure of [2]cat-GFP predicted by AF2. Two rings constituting [2]cat-GFP are colored red and green, respectively. **g** Structure of [3]cat-GFP predicted by AF2. Two rings constituting [3]cat-GFP are colored red and green, respectively. **h** LC-MS spectra of [2]cat-GFP and [3]cat-GFP. **i** LC-MS spectra of cat-mCherry (Two rings constituting cat-mCherry are colored red and pink, respectively), cat-mWasabi (Two rings constituting cat-mWasabi are colored red and cyan, respectively), and cat-YPet (Two rings constituting cat-YPet are colored red and yellow, respectively). Source data are provided as a Source Data file.

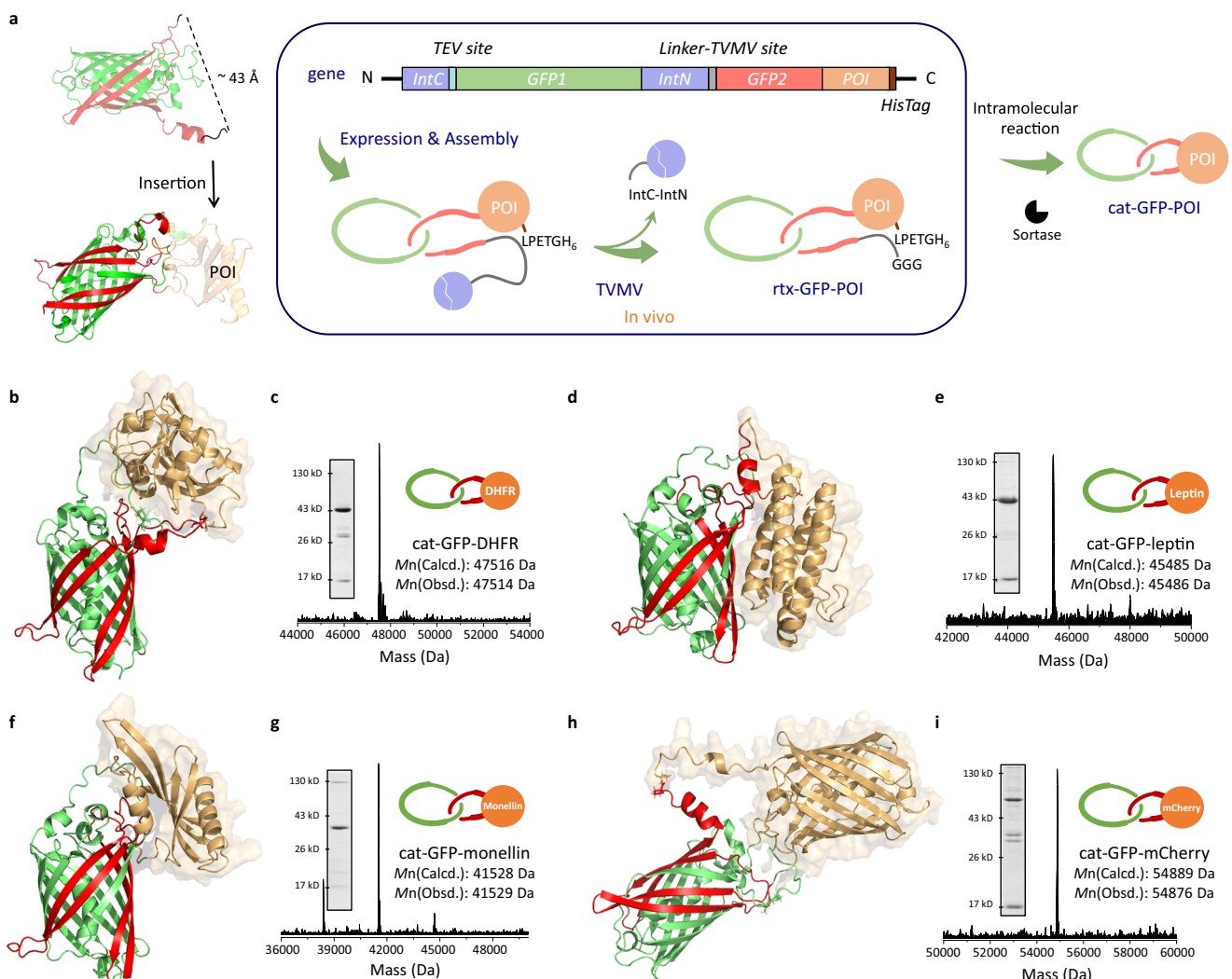

**Fig. 4 | cat-GFP as a robust scaffold for fusion protein catenanes. a** Synthetic strategy of fusion protein catenanes. Structure, LC-MS spectra, and SDS-PAGE analysis of cat-GFP-DHFR (**b**, **c**), cat-GFP-leptin (**d**, **e**), cat-GFP-monellin (**f**, **g**), and cat-GFP-mCherry (**h**, **i**). The structures of cat-GFP-POI were predicted by AF2. Two components from GFP are colored red and green, protein of interest (POI) is colored orange. Source data are provided as a Source Data file.

hindrance of the POI promoting intra-complex reaction. The TEV digestion experiments further verify their catenane topology (Supplementary Fig. 10). The fluorescence spectra of fusion protein catenanes suggested the effective chromophore maturation (Supplementary Fig. 11). Their structures were also predicted by AF2 (Fig. 4b, d, f, and h), from which two mechanically interlocked rings (GFP1: green; GFP2-POI: red-orange) could be clearly discerned. These results indicate that fusion protein [2]catenanes could be selectively obtained based on the cat-GFP scaffold upon POI insertion.

### Effects of mechanical interlocking

Previously, it has been reported that catenation could enhance protein stability due to conformational restriction and increase in effective concentration[46,47]. Catenation has also been reported to improve protein's mechanical resilience against freeze-and-thaw cycles and enhance enzymatic activity by the multivalent synergy between two subunits[26]. Herein, to illustrate the topological effect of catenation, we designed both linear and cyclic proteins with almost identical amino acid composition to catenane as topological controls, namely l-GFP(-POI) and c-GFP(-POI). They possess molecular compositions and overall folded structures almost identical to the corresponding catenanes, as evidenced by the LC-MS spectra and SEC overlay. The fluorescence spectra of all protein samples again confirm the

chromophore maturation with the expected loss of a water molecule and two hydrogen atoms corresponding to a difference of 20 Da in molecular weight (Supplementary Figs. 12–16).

The l-GFP, c-GFP, and [2]cat-GFP show nearly identical elution profiles in SEC (Fig. 5a), suggesting a very similar overall structure in solution. Their melting points ($T_m$) measured by differential scanning calorimetry are 95.4 °C, 95.4 °C, and 90.9 °C, respectively (Table 1 and Supplementary Fig. 17). Cyclization does not impact the $T_m$, but splitting it into two components results in a lower $T_m$ in cat-GFP. This is probably because the entanglement was mainly introduced into the GFP domain on the loop region, which does not span the hydrophobic core and is somewhat superficial. On the other hand, thermal resilience is an important parameter characterizing dynamic stability and represents the protein's capacity to refold to its functional conformation after denaturation[48]. We envisioned that due to the constrained motion of two rings, catenane should exhibit excellent thermal resilience. We thus boiled these samples for 4.5 min and then cooled down to 25.0 °C following a programmed protocol during which the fluorescence intensity was constantly monitored (Fig. 5b)[49,50]. Indeed, the fluorescent recovery ratio (normalized by original fluorescence) shown in Fig. 5c indicates cat-GFP has a sharp slope and nearly 100% recovery of fluorescence within 50 min, c-GFP has a less steep recovery curve with a final recovery ratio of ~70%, and

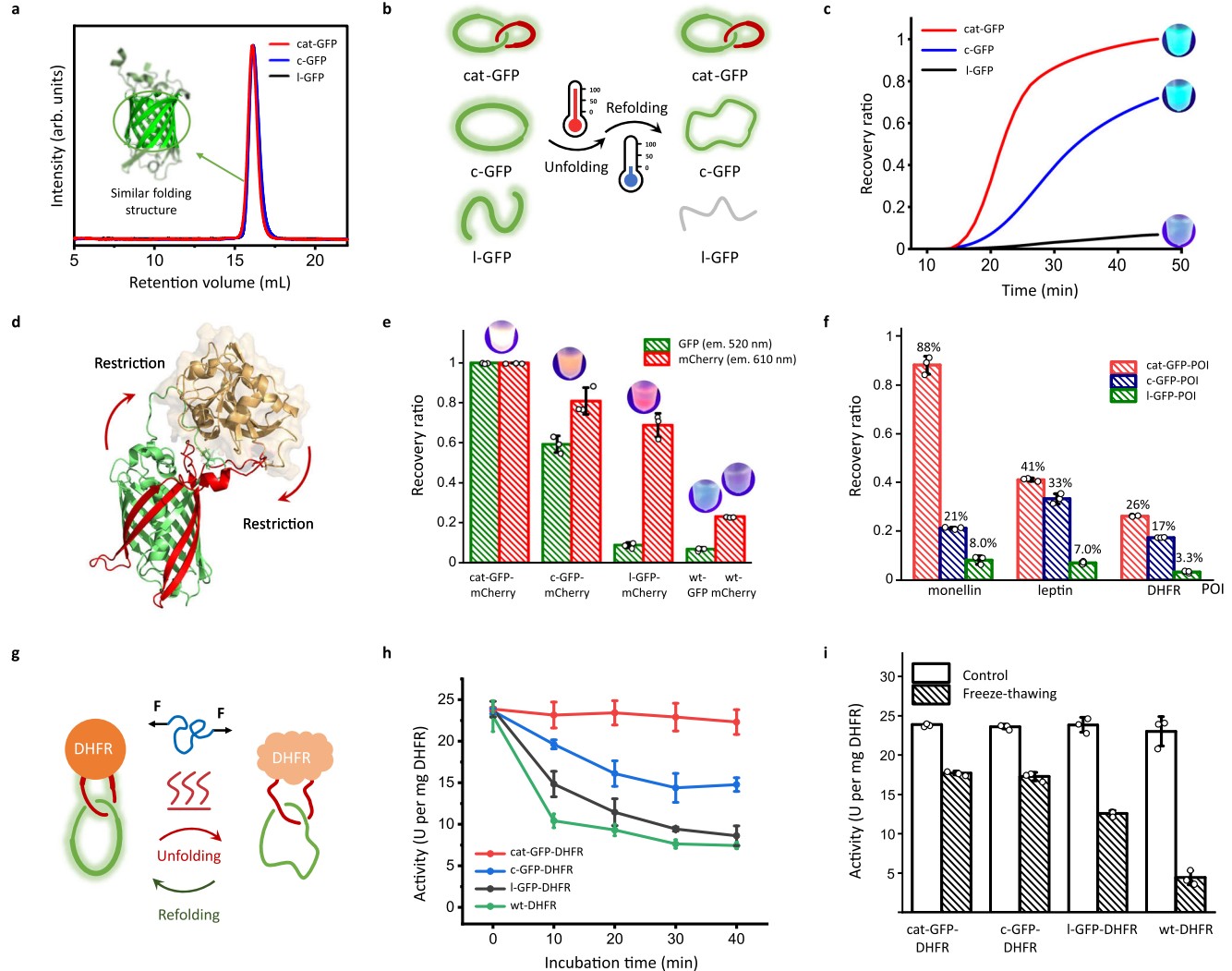

**Fig. 5 | Influence of catenation on protein properties. a** SEC overlay of cat-GFP (red), c-GFP (blue), and l-GFP (black), showing their similar folding structures. **b** Process for evaluating the recovery capability of GFP samples. The samples were heated to 99.9 °C and subsequently cooled down to 25.0 °C for refolding. **c** Fluorescence recovery curves of cat-GFP (red), c-GFP (blue), and l-GFP (black). **d** Catenation through cat-GFP scaffold may convey functional benefits to the enclosed POI through increasing the effective concentration of ring components and exerting entropy restriction. **e** Fluorescence recovery ratio of cat-GFP-mCherry, c-GFP-mCherry, l-GFP-mCherry. wt-GFP, and wt-mCherry in 520 nm channel (green) and 610 nm channel (red). **f** Fluorescence recovery ratio of cat-GFP-

POI (red), c-GFP-POI (green), and l-GFP-POI (blue), with different POIs (monellin, leptin, and DHFR) inserted. **g** cat-GFP scaffold shows the potential of enhancing POI's capacity to renature after thermal and mechanical denaturation. **h** Catalytic activities of cat-GFP-DHFR (red), c-GFP-DHFR (blue), l-GFP-DHFR (black), and wt-DHFR (green) after annealing at 50 °C for designated times. **i** Catalytic activities of cat-GFP-DHFR, c-GFP-DHFR, l-GFP-DHFR, and wt-DHFR before (blank) and after (sparse) freeze-thawing. Data points are presented as mean values of triplicates ($n = 3$ independent experiments) with ± standard deviation shown as error bars in Fig. 5e, f, h, i.

l-GFP hardly refolds within the same period of time (to 7%). The distinct recovery profiles demonstrate the subtle and significant effects brought by topology. Due to the constrained motion of two rings, protein catenanes tend to show better capacity for refolding after denaturation than their linear or circular counterparts. The unfolded cat-GFP perhaps samples a much smaller conformational space than both linear and cyclic controls during the synergistic refolding of two rings.

## Mechanical coupling between interlocked subunits
The fusion catenane samples each exhibit two melting peaks in DSC, indicating that the unfolding events of GFP and POI are relatively independent (Table 1 and Supplementary Fig. 18). A common peak attributed to GFP unfolding was observed for all constructs with a characteristic melting point ($T_{m, GFP}$) of ~96 °C for l-GFP-POI, ~95 °C for c-GFP-POI, and ~91 °C for cat-GFP-POI. The trend is similar to that

without POI. Each POI gave a second unfolding peak with another melting point ($T_{m, POI}$). For DHFR, the $T_m$ in the linear construct (50.7 °C) is similar to that of the wild-type, but increased to 55.9 °C in the cyclic construct and further to 57.2 °C in the catenane construct. For monellin, the $T_m$ is almost the same in l-GFP-monellin (69.9 °C) and c-GFP-monellin (70.1 °C), showing a moderate increase relative to the wild-type (66.4 °C), but further increases to 73.1 °C in cat-GFP-monellin. We attributed this change to the conformational restriction on the POI exerted by the GFP domain ($T_m$ ~ 91 °C). Though absent in the linear construct, the unfolding events of GFP and POI are partially coupled together in the cyclic construct and even more so upon catenation[51]. Compared with simple cyclization, catenation could impose more conformational restriction on the protein domain, thus enhancing the stability of POIs to a larger extent. In particular, mCherry and GFP are of comparable stability and their unfolding events in the catenane construct become strongly coupled (Supplementary

**Table 1 | Thermal stability of GFP-POIs**

| Proteins | $T_{m, GFP}$/°C | $T_{m, POI}$/°C |
|---|---|---|
| l-GFP | 95.4 | \ |
| c-GFP | 95.4 | \ |
| [2]cat-GFP | 90.9 | \ |
| wt-DHFR | \ | 53.4 |
| l-GFP-DHFR | 95.5 | 50.7 |
| c-GFP-DHFR | 95.0 | 55.9 |
| cat-GFP-DHFR | 90.5 | 57.2 |
| wt-monellin | \ | 66.4 |
| l-GFP-monellin | 95.6 | 69.9 |
| c-GFP-monellin | 95.3 | 70.1 |
| cat-GFP-monellin | 90.4 | 73.1 |

$T_m$ values of cat-GFP(-POI), c-GFP(-POI), l-GFP(-POI), and single-domain POI. According to the official guide of MicroCal PEAQ-DSC, the measurement reproducibility gives an error of <0.18 °C and the system reproducibility gives an error of <0.1 °C. Therefore, the total error of our experiment would be <0.21 °C according to the error formula.

Table 2). Both linear and cyclic fusion exhibit a sharp melting peak with a similar $T_m$ to l-GFP, while the catenane fusion shows a much broader melting peak with a lower $T_m$ similar to cat-GFP.

We also evaluated the thermal resilience of these fusion samples by fluorescence recovery. All cat-GFP-POIs showed better fluorescence recovery compared to the corresponding linear and cyclic controls, suggesting that the functional benefits of single-domain catenation could be retained upon POI insertion (Fig. 5f) despite the slightly lower $T_m$ of cat-GFP. To assess the changes in POI, we first simultaneously monitored the fluorescent recovery of GFP with the emission maximum at 520 nm and mCherry at 610 nm[52]. It was found that both domains have almost complete fluorescence recovery in the catenane construct, which is superior to the linear and cyclic controls, as well as the wild-type GFP or mCherry (Fig. 5e and Supplementary Fig. 19). Notably, the fluorescence recovery performance of mCherry was generally better than GFP, consistent with the trend in the wild-type samples.

## Functional benefits of fusion catenanes

Thermal resilience is crucial for industrial enzymes[53]. It is natural to investigate whether the thermal resilience of cat-GFP has been conferred onto the POIs. Hence, we measured the activity of DHFR in these fusions after treating the samples first by incubating at 50 °C for a certain time followed by cooling down to 4 °C overnight. As shown in Fig. 5h, the catalytic activity of each sample decreased with increasing incubation time, among which cat-GFP-DHFR displayed the steadiest performance with fully retained activity over prolonged incubation and wt-DHFR lost ~60% of its initial activity after 40 min of incubation (Fig. 5h and Supplementary Fig. 20). Both l-GFP-DHFR and c-GFP-DHFR retained a better catalytic activity than wt-DHFR, yet inferior to that of the catenane construct. It indicates that while GFP fusion could in general enhance thermal resilience, catenation turns out to be the most effective. We also investigated the samples' mechanical resilience against freeze-and-thaw-induced denaturation. This is another important factor affecting the shelf-life of therapeutic proteins and enzymes[54]. The retained activity ratios were comparable for cat-GFP-DHFR and c-GFP-DHFR at >75%, but the value fell to 50% in l-GFP-DHFR and 19% in wt-DHFR, showcasing the power of topology engineering (Fig. 5i and Supplementary Fig. 21). Overall, the cat-GFP scaffold endows the protein better thermal and mechanical resilience compared to their linear and cyclic counterparts.

## Cellular synthesis of cat-GFP

The two-step approach offers better control over the synthetic process. Not only can the side products be removed during each step, but it also affords intriguing products like [3]cat-GFP. Nevertheless, it still takes extra steps of in vitro manipulation and is time-consuming. It is highly desired to move the entire process into cells, provided that the assembly and reaction could act in concert to give the target catenane selectively. Building on our previously reported streamlined synthesis method[22], we designed a gene cassette of *IntC1-GFP1-IntN1-IntC2-GFP2-IntN2* (Supplementary sequence 4) where genes encoding *GFP1* and *GFP2* are inserted between gene encoding two orthogonal intein pairs from *Nostoc punctiforme* (Npu)[55]. Direct cellular synthesis gives the desired single-domain catenane along with considerable amounts of the side-product composed of two "unlinked" rings. The purity of the catenane was merely ~34% (Fig. 6b and Supplementary Fig. 22). This noncovalent complex is a true isomer to cat-GFP, yet without entanglement. It is stable under native conditions and difficult to separate further. Interestingly, very few [3]catenanes were obtained in contrary to that of in vitro experiments, which is probably due to the limited cellular concentration as a result of slow expression and fast reaction.

To delay ring closure and ensure sufficient time for entanglement formation, we switched to split VidaL inteins[56] with slower trans-splicing kinetics and orthogonal reactivity to split Npu inteins for optimizing the synthetic selectivity. The gene cassette of *IntC1-GFP1-IntN1-VidC-GFP2-VidN* was constructed and expressed to give cat-GFP. The crude product contains higher amounts of cat-GFP (~45%), indicating a better match between reconstitution and the trans-splicing (Fig. 6b, Lane 2 and Supplementary Fig. 22). Nevertheless, it remains difficult to purify cat-GFP from the "un-link" under native conditions (Supplementary Fig. 23), despite many attempts. Eventually, thermal incubation turned out to work well, taking advantage of the difference in thermal stability of cat-GFP and the "un-linked" isomer. Incubation at 75 °C for 3 h leads to extensive aggregation of c-GFP1, which could be removed by high-speed centrifugation. The smaller fragment c-GFP2 in the supernatant was easily removed by dialysis with a molecular weight cutoff of 15 kDa (Fig. 6b), improving the catenane purity to ~87% (Fig. 6b, Lane 4 and Supplementary Fig. 22). The LC-MS spectrum verifies the expected molecular weight of cat-GFP (Fig. 6c). TEV digestion experiments were performed and the products were characterized by SDS-PAGE and LC-MS to verify the topology. Moreover, the fluorescence spectrum of VidaL-cat-GFP unambiguously confirmed the chromophore maturation (Supplementary Fig. 24). Therefore, cat-GFP can be conveniently obtained through direct cellular synthesis and thermal purification. We have attempted to apply this method to the synthesis of cat-GFP-POIs. Unfortunately, the folding of GFP and POI were hardly in synergy and most fusions were obtained as a single ring. Hence, it is actually nontrivial to achieve the "assembly-reaction" synergy in cellulo.

## Folds of topological proteins

Linear chains are privileged in their ability to fold into different shapes, typically without entanglements[57]. Intriguing questions thus arise: "Are there any protein universes that are based entirely on nonlinear, topological scaffolds like catenanes? If so, how to traverse into such parallel protein universes and what kind of property changes do we anticipate?" In this work, we show that nonlinear chains with unconventional topology may also fold synergistically into a single domain with strong conformational coupling between subunits. The strategy shall work for other protein folds as well. In other words, theoretically, the current linear protein universe can be mapped into multiple parallel protein universes, each based on a unique type of backbone chain topology. Notably, even just for the Hopf Link topology, there could be many different ways to construct a single-domain protein catenane from a common precursor. Therefore, it provides access to protein folds in the parallel protein universe, representing entirely new territories in macromolecular science. Although direct evolution may be required to further adapt these sequences into the new fold, the possibility to combine the topological effects on stabilization and the rich

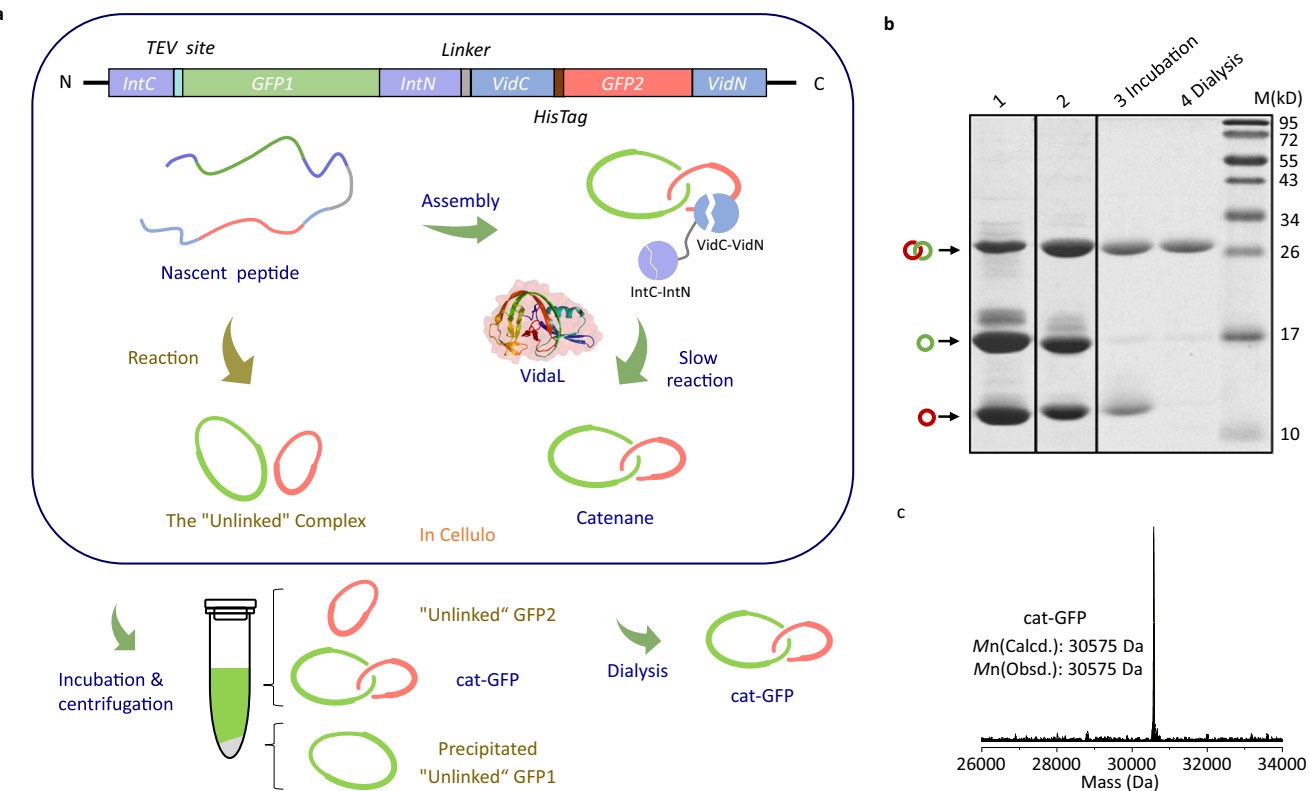

**Fig. 6 | Cellular synthesis and purification of cat-GFP. a** Schematic representation for cellular synthesis and purification of cat-GFP. **b** SDS-PAGE analysis of the crude products obtained by Npu intein (lane 1, ~34% yield for cat-GFP), by Npu and VidaL intein (lane 2, ~45% yield for cat-GFP), and purified after incubation at 75 °C for 3 h (lane 3, ~87% yield for cat-GFP) and further by dialysis (lane 4). **c** LC-MS spectrum of the purified cat-GFP obtain by cellular synthesis. Two rings constituting cat-GFP are colored red and green, respectively. Source data are provided as a Source Data file.

functions of protein is an enticing goal in affording unique solutions in protein engineering[58].

## Discussion

In summary, we have developed a general approach to construct a single-domain green fluorescent protein catenane as the counterpart of the GFP fold in the linear protein universe. The entanglement was introduced via rewiring the connectivity of the GFP precursor and the synthesis was achieved via either a two-step templated synthesis or assembly-reaction synergy in cellulo. The method works for other proteins with similar fold, giving rise to a family of single-domain protein catenanes like cat-YPet, cat-mCherry, and cat-Wasabi. Meanwhile, it also provides a robust scaffold to make fusion catenanes with POIs inserted at the loop region. In comparison to the linear and cyclic controls, cat-GFP(-POI) shows the best thermal resilience. Due to conformational restriction, the embedded POIs exhibit improved thermal stability, thermal resilience, and mechanical stability. The results suggest that topology could bring in significant changes in properties. Beyond the current linear protein universe, there may exist multiple parallel protein universes based on topological protein scaffolds like catenanes, probably with desirable functional benefits. Traversing into these new territories is expected to bring in additional functional benefits, such as thermal resilience, for proteins. The current work makes these territories accessible in practice and open for exploration. These new topological protein folds are probably highly evolvable. This work also reveals the primary effects of topology on structure-property relationship. Continued endeavors are being devoted to further promoting the development of therapeutic proteins and industrial enzymes based on topological proteins toward a bright future in application.

## Methods

### DNA construction

All oligonucleotide primers were ordered from Genewiz Inc, SZ, and their sequences are provided in Supplementary Data. The genes of GFP-OPT, sf-mCherry2, monellin, and VidaL intein were ordered from Shanghai Generay Biotech Co., Ltd. Sequences encoding mWasabi, YPet, leptin, DHFR, and Npu intein were amplified by PCR from the corresponding plasmids previously constructed in our lab. *IntC-GFP1-IntN-GFP2*, *IntC-YPet1-IntN-YPet2*, *IntC-mCherry1-IntN-mCherry2*, *IntC-mWasabi1-IntN-mWasabi2*, *IntC-GFP1-IntN-GFP2-POI*, *IntC1-GFP1-IntN1-IntC2-GFP2-IntN2*, and *IntC1-GFP1-IntN1-VidC-GFP2-VidN* were cloned into the bacterial co-expression vectors pMCSG19/pRK1037, and other genes were cloned into pET-15b or pET-22b vectors by standard restriction digestion and ligation protocols. All DNA sequences were confirmed by direct sequencing.

### Protein expression and purification

For in situ protease cleavage, the plasmids based on the pMCSG19 vector were introduced into the *E. coli* BL21(DE3) competent cells containing the pRK1037 vector. The single colony was cultured on 2xYT plates containing 100 μg/mL ampicillin and 50 μg/mL kanamycin, and then inoculated into 5 mL of 2xYT broth with the same antibiotics and grown in a shaker (37 °C, 250 rpm). The overnight cultures were inoculated to 300 mL of fresh 2xYT broth containing 100 μg/mL ampicillin and 50 μg/mL kanamycin. When $OD_{600}$ reached 0.8-1.3, isopropyl-β-D thiogalactopyranoside (IPTG) was added to a concentration of 0.5 mM to induce target protein expression. Then cultures were shaken at 16 °C for 12 h. The other plasmids containing the designed sequences were transformed into BL21(DE3) for expression. Cells were harvested by centrifugation (5500 g × 15 min, 4 °C). The harvested cell pellets were resuspended in 20−30 mL lysis buffer

(50 mM NaH$_2$PO$_4$, 300 mM NaCl, 10 mM imidazole, pH 8.0), and lysed by ultrasonication. The supernatant was then collected after centrifugation (12,000 g × 30 min, 4 °C), and mixed with Ni-NTA resin (GE Healthcare. Inc.). The mixture was incubated at 4 °C for 1 h, then loaded to an empty column and washed by wash buffer (50 mM NaH$_2$PO$_4$, 300 mM NaCl, 20 mM imidazole, pH 8.0) for five resin volumes, finally eluted by elution buffer (50 mM NaH$_2$PO$_4$, 300 mM NaCl, 250 mM imidazole, pH 8.0). The eluted products were collected and further purified by size exclusion chromatography (SEC), which was performed on a Superdex 200 increase 10/300 GL column in an ÄKTA FPLC system (GE Healthcare, Inc.) using TN buffer (150 mM NaCl, 20 mM Tris-HCl, pH 8.0) as the mobile phase at a flow rate of 0.5 mL/min. Protein concentrations were determined by UV absorbance with NanoPhotometer P330 (Implen, Inc.).

### Protein characterization

Sodium dodecyl sulfate-polyacrylamide gel electrophoresis (SDS-PAGE) was performed to analyze the protein samples after being mixed with 5x SDS-PAGE loading buffer (250 mM Tris-HCl, 50% glycerol, 10% SDS, 250 mM β-mercaptoethanol, 0.05% bromophenol blue) and heated at 98 °C for 10 min. Relative protein quantification was performed using Image Lab Software (Bio-Rad) for SDS-PAGE images. Ultra-performance liquid chromatography-mass spectrometry with quadrupole rods SQ Detector 2 mass spectrometer (Water Corp.) was utilized to confirm the molecular weights of all protein samples.

### Formation of single-domain GFP catenanes

Sortase-mediated ligation was conducted according to the protocols reported before[27,59]. Briefly, protein samples were diluted to 50 μM, then sortase A (0.1 equivalent) and calcium chloride (10 mM) were added to initiate sortase A-mediated cyclization. The reaction mixture was incubated at 16 °C for at least 3 h. The crude products were obtained by loading the reaction mixture to a Ni-NTA column and collecting the flow-through and wash components using wash buffer (20 mM Tris-HCl, 500 mM NaCl, 20 mM imidazole, pH 8.0). The unreacted component with His-Tag as well as sortase A having a higher affinity with Ni-NTA were recovered by 250 mM imidazole. Protein catenanes were finally obtained from the crude products by another round of SEC purification.

### TEV protease cleavage

The protein solution was mixed with TEV protease at a molar ratio of 20:1. Proteolysis was carried out at 30 °C for at least 10 h and quenched by adding 5x SDS-PAGE loading buffer and heating at 98 °C for 10 min. The resulting products were analyzed by SDS-PAGE and LC-MS.

### Fluorescence spectroscopy

For excitation and emission spectrum screening, protein samples in TN buffer (pH = 8.0) were added to the black microplate, and fluorescence was monitored using EnSpire multimode plate reader (PerkinElmer Inc.) by scanning mode. All normalized data were normalized according to the maximum values by Origin2020b.

### Heat treatment and detection of fluorescence recovery

The recovered activity of GFP was evaluated by measuring fluorescence intensity changes over time during programmed cooling process. 10 μL 1 mg/mL protein samples were added to opaque white 8-tube strips and centrifuged to the bottom. The change of fluorescence was recorded by Real-Time PCR (RT-PCR) Systems (StepOne-Plus Real-Time PCR System, Applied Biosystems) using the channel for FAM reporter (470 nm/520 nm) or ROX reporter (470 nm/610 nm). Experimental data points and error bars are presented as mean ± standard deviation of a minimum of three replicates. The programmed temperature change protocols are shown in Supplementary Fig. 25.

### Differential scanning calorimetry test

1 mg/mL samples were tested using MicroCal PEAQ-DSC (Malvern Panalytical Ltd.). Samples were scanned every 2 °C per minute for each sample. Data were processed using MicroCal PEAQ DSC software.

### DHFR catalytic activity assay

Protein samples were diluted to 100 nM with KHP buffer (50 mM potassium phosphate-containing 80.2% K$_2$HPO$_4$ and 19.8% NaH$_2$PO$_4$, 5 mM β-mercaptoethanol, pH = 7.4). 10 μL of 100 nM protein samples were then mixed with 40 μL of 0.5 mM NADPH (in KHP buffer, sigma) and 60 μL of 0.33 mM dihydrofolic acid (in KHP buffer, sigma) in 96 microwell plate, and the absorbance at 340 nm was measured immediately in kinetic mode at 25 °C using EnSpire multimode plate reader (PerkinElmer Inc.). The absorbance of NADPH with varying amounts (10, 20, 30, 40, 50, 60, 100 nmol) at 340 nm was also measured to get the standard curve, which was used to calculate the amounts of NADPH remained during the reaction. A linear range of samples plot was used to calculate the DHFR activity. For thermal resilience test, work solutions of samples were heated at 50 °C for 10 min, 20 min, 30 min, or 40 min and then put into 4 °C immediately to cool down. The freeze-thawing samples were first frozen at −80 °C for several days and then thawed at 25 °C. The activities were tested as the methods described above. Experimental data points and error bars are presented as mean ± standard deviation of a minimum of three replicates.

### Protein visualization

PyMOL-2.4.0[60] was used for visualization and analysis.

### Statistics and reproducibility

Synthesis of cat-GFP, cat-FPs, cat-GFP-POIs, and VidaL-cat-GFP were repeated at least three times, and similar SDS-PAGE were obtained with each batch (Figs. 3b, 4c, e, g, i, 6b, and Supplementary Figs. 9b, e, h, k). TEV protease cleavage experiments of cat-GFP, cat-FPs, cat-GFP-POIs and VidaL-cat-GFP (Fig. 3e, Supplementary Figs. 7a, 10a, 10c, 10e, 10g, 24a) were repeated at least two times with similar results. The purification of rtx-GFP and VidaL-cat-GFP (Fig. 6b and Supplementary Figs. 3a, 3c, 23c) were repeated at least three times with similar results. The cellular synthesis of Design A, Design B, and Design C were repeated at least two times (Supplementary Fig. 2k).

### Reporting summary

Further information on research design is available in the Nature Portfolio Reporting Summary linked to this article.

## Data availability

The data that support the findings of this study are available within the paper and its Supplementary Information files. The structure of GFP used in this study are available in the Protein Data Bank, PDB ID: 2B3P [rcsb.org/structure/2b3p]. Source data are provided with this paper.

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

## Acknowledgements

We are grateful for the financial support from the National Natural Science Foundation of China (Nos. 21991132, 21925102, 92056118, 22101010, 22201016, 22201017), the National Key R&D Program of China (No. 2020YFA0908100, 2020AAA0105200), and Beijing National Laboratory for Molecular Sciences (BNLMS-CXXM-202006). W.-B. Z. acknowledges Bayer Pharma for the Bayer Investigator Award.

## Author contributions

W.-B.Z. conceived the project. W.-B. Z., Z.Q., and J.F. designed the experiments. Z.Q. performed the experiments and collected the data. W.-B.Z. and Z.Q. analyzed the data with inputs from J.F., and Y.-X.W. W.-B.Z., Z.Q., Y.S., Y.L., and W.-H.W. wrote the manuscript. All authors discussed and commented on the paper.

## Competing interests

W.B.Z. and Z.Q. are authors on a patent application in China regarding the construction of a single-domain fluorescent protein catenane and its application in the synthesis and application of fusion protein catenanes (CN 202211561433.5). All other authors declare no competing interests.
