## [Peer Review File · Nature Communications]

Reviewers' Comments:

Reviewer #1:

Remarks to the Author:

Qu and colleagues report on the design, assembly and characterization of a GFP catenane. The researchers also use the catenane approach to circularize other proteins and characterise their stability.

It is an interesting idea to explore proteins with different topologies and GFP is a good target to demonstrate catenanes are possible: extensive circular permutation and split-expression have been reported with GFP, leaving the authors to focus on the threading of the two circular peptides. I think it is premature for the authors to claim that this will "bring benefits indispensable for functional proteins (p.19, L331)", as this was not demonstrated in this manuscript.

I think some points should be addressed before publication:

1. Make it more prominent how efficient the assemblies (both in vitro and in vivo) are. The only place where I could see how robust the system was, was on SI Figure 21 where it is finally stated that catenanes were being expressed around 19 mg/L, with efficiencies of assembly above 80% (in some conditions).
2. There is significant characterisation of cat-GFP-POI but the text keeps emphasising how this approach is improving POI stability. However, how much stabilization could be imparted to the POIs by simply circularizing them (a subtler change in topology and a well-established route towards greater thermal stability)? Or cyclizing a GFP-POI fusion, which would maintain GFP/POI close interaction and potential local thermal stabilization? As is, it is unclear what the "unique selling point" of the catenanes is.
3. Molecular weight markers are often not displayed on gels (Figure 3 and Figure 4). Please include them. If they are being omitted due to space or stylistic constraints, the full gels can be submitted as SI figures.

I also spotted a couple of minor points:

- * P4.L73 - 1,062,864 is $\sim 10^6$
- * A couple of places - *facilely*. While technically correct, this word is rarely used in British English. Consider replacing it with "easily", which is also more accessible.
- * P16.L274 - "Inspired by the reported streamlined synthesis method" feels like an inappropriate way to cite a publication that shares 3 authors with the current manuscript. I think that "Building on our reported streamlined synthesis method" may be more appropriate.

Reviewer #2:

Remarks to the Author:

In this manuscript, Zhang and coworkers describe strategies to form single domain GFP catenanes and their successful implementation. Key to the formation of GFP catenanes is the use of a rewiring process to reconnect GFP's secondary motifs. The formation and characterization of resulting protein constructs were verified by SDS-PAGE analysis and size exclusion chromatography. Overall, this is an interesting demonstration of single domain GFP catenanes and publication of this work in Nature Communications is recommended. The authors are advised to address the following issues before this paper is published.

1. In Fig. 3, when the catenane formation provides multiple products, an estimation of product selectivity will be useful. The presence of fluorescent motif in these catenane constructs can potentially make this quantification easier, but other simpler assays are also available.
2. Further exploring the applications of these intriguing protein catenanes would be of interests to the community.

Reviewer #3:

Remarks to the Author:

The work from Zhiyu Qu very nicely describes the approach towards the creation of relict free single domain catenanes. To exemplify the approach, they choose the green fluorescent protein (here superfolder variant sfGFP). The manuscript is well written and consistently tells the story of the catenanes creation as well as their advantages. While the authors eventually fail to fully produce the catenane in vivo with high yield they nonetheless provide a number of insightful aspects to this field. The work is in general experimentally sound and with sufficient experimental controls.

Nonetheless there are a number of major aspects which I would like the authors to touch upon.

1) The following is a general writing almost philosophical aspect. While I appreciate the charm of speaking of "parallel fold universes" ("that the linear proteins family can be mapped catenane protein family") with the single current paper (and also the small body of previous work) I do not know if this wording is really justified. It suggests that each protein can be relatively straightforward converted into a catenate twin. Secondly, the wording in the abstract of "current protein universe" when speaking about the linear protein world suggests clear evidence that another fold universe with the same players was preceding or is about to come. From my perspective this is not backed by current knowledge.

Either the authors need to back up even their "questions" (l. 34 ff.) with ample literature reviewing from the field of protein / fold evolution or should rather cut this whole notion from the abstract and introduction and discuss it in the discussion section of the manuscript. Beyond that while "design principles" are mentioned (l. 66 ff.) those are very vague and general and do not constitute a claim to generalization. Also, such a claim would require additional fold classes to be tackled with this method. Thus, given that the example chosen by the authors uses a protein with known reconstitution (Cabantous, 2005) I would like the authors to reduce this notion of generalization.

2) The comparison to wt sfGFP is lacking a bit. While I see that a good and direct comparison is the I-GFP the authors modeled after the sequence of their catenane versions I am not sure how well the catenanes fare in comparison to a "native" protein. The authors mention wt comparisons only once regarding the refolding (Fig. 5 and Suppl. Fig. 18). Especially with this data I am quite surprised on the described low refolding of their wt based on the superfolder, since literature (Andrews, J Mol Biol , 2009, GdnHCl, though, Temp needs to be looked up) indicates almost complete refolding for the folder optimized versions of avGFP. The limited recovery of around 10% the authors state appears very low. I would ask the author to revisit that data and also add the broad literature on refolding studies of FPs to their text to put the behaviors they study in context.

3) The authors do not comment much on the chromophore maturation. All spectral characterizations are normalized to the peak. I would like to see raw absorbance spectra including the 280 nm protein peak of all described constructs in comparison to wildtype to judge the effectivity of chromophore maturation not only protein folding.

4) The authors claim additive stability effects of catenane based fusions. Since chimeric proteins are not limited to catenanes but often challenging it would be interesting to compare at least one of their chimeric constructs against a linear fusion based on a wt GFP where loop insertions are equally possible to judge the net effect of this approach.

5) The enzymatic activity test has only been done on one protein of interest fusion (DHFR). Have the others not been tried (why?) or do they give unsatisfactory results.

6) The last major point is more out of interest. Chimeras on FPs are widely applied in sensor engineering (e.g. GCaMPs) for this the fusion to the protein(s) of interest is done in beta sheet 7 near the chromophore. Would such also work with the catenanes, could it be tried and might it add additional advantages to creating photoactive chimeras of FPs?

Minor points:

I. 78: what does "representative" 10 designs vs. "we designed many" mean

I .98: what does "Among numerous trials," mean. Pls. specify and also inform on unsuccessful trials and their properties

Supp Fig 18f: Difference between legend and figure (black and green)

Most figures: Error bars are not commented on. Are those from independent productions of the protein and then measurement (recommended) or repeated measurements?

Wen-Bin Zhang, Ph.D. Professor
College of Chemistry and Molecular Engineering,
Center for Soft Matter Science and Engineering
Peking University, Beijing, 100871, P. R. China
Tel.: +86-62766876; E-mail: wenbin@pku.edu.cn

April 13, 2023

Dear Dr. Cloney,

Thank you very much for handling our manuscript. We also thank the reviewers for their thoughtful comments and suggestions, which have helped improve the quality of our paper. Enclosed please kindly find the revised manuscript and the one with major changes highlighted. The point-by-point responses are listed below:

Reviewer #1

“Qu and colleagues report on the design, assembly and characterization of a GFP catenane. The researchers also use the catenane approach to circularize other proteins and characterise their stability.

It is an interesting idea to explore proteins with different topologies and GFP is a good target to demonstrate catenanes are possible: extensive circular permutation and split-expression have been reported with GFP, leaving the authors to focus on the threading of the two circular peptides. I think it is premature for the authors to claim that this will “bring benefits indispensable for functional proteins (p.19, L331)”, as this was not demonstrated in this manuscript.”

Reply: We thank the reviewer for the suggestion. We have rephrased the sentence on page 19, and “bring benefits indispensable for functional proteins” is changed into “bring in additional functional benefits, such as thermal resilience, for proteins”.

“1. Make it more prominent how efficient the assemblies (both in vitro and in vivo) are. The only place where I could see how robust the system was, was on SI Figure 21 where it is finally stated that catenates were being expressed around 19 mg/L, with efficiencies of assembly above 80% (in some conditions).”

Reply: We thank the reviewer for the suggestion. We have added “yielding 83% protein catenane (46% for [2]cat-GFP and 37% for [3]cat-GFP) after Ni-NTA purification” on page 7 and added “showing the yields of in vitro synthesis (~46% [2]cat-GFP and ~37% [3]cat-GFP, determined by gel densitometry)” in the caption of Fig.3b to emphasize the efficiency of the in vitro synthesis. The yields have also been added to the caption of Fig. 6b to show the efficiency of the in vivo synthesis.

“2. There is significant characterisation of cat-GFP-POI but the text keeps emphasising how this approach is improving POI stability. However, how much stabilization could be imparted to the POIs by simply circularizing them (a subtler change in topology and a well-established route towards greater thermal stability)? Or cyclizing a GFP-POI fusion, which would maintain GFP/POI close interaction and potential local thermal stabilization? As is, it is unclear what the "unique selling point" of the catenanes is.”

Reply: We thank the reviewer for the critical comments. Simply circularizing them could also enhance the stability of POIs, but to a less significant extent than catenation. We have constructed the controls by simply cyclizing a GFP-POI fusion. The degree of stabilization imparted to the POIs by circularization or catenation is summarized in Table 1. For DHFR, the T_m in the linear construct (50.7 °C) is similar to that of the wild-type but increased to 55.9 °C in the cyclic construct and further to 57.2 °C in the catenane construct. A similar tendency was also observed in monellin, the T_m is almost the same in *l*-GFP-monellin (69.9 °C) and *c*-GFP-monellin (70.1 °C), showing a moderate increase relative to the wild-type (66.4 °C), but further increase to 73.1 °C in *cat*-GFP-monellin. We attributed this change to the conformational restriction on the POI exerted by the GFP domain ($T_m \sim 91$ °C). Compared with simple cyclization, catenation with two cyclized components interlocking each other could impose more conformational restriction on the protein domain and thus enhance the stability of POIs to a larger extent.

Besides stability, catenation also enhanced the thermal resilience much more significantly than simple cyclization (Fig. 5c, 5e, and 5f). All *cat*-GFP(-POIs) showed better fluorescence recovery compared to the corresponding linear and cyclic controls, suggesting the functional benefits of single-domain catenation could be retained upon POI insertion. Due to the constrained motion of two rings, protein catenanes tend to show better capacity for refolding after denaturation than their linear or circular counterparts.

To clarify the functional benefits of catenanes, we have added “due to the constrained motion of two rings, protein catenane showed a better capacity of refolding after denaturation than linear or simply cyclized protein” on page 12 and “Compared with simple cyclization, catenation with two cyclized components interlocking each other could impose more conformational restriction on the protein domain and thus enhanced the stability of POIs to a larger extent.” on page 14.

“3. Molecular weight markers are often not displayed on gels (Figure 3 and Figure 4). Please include them. If they are being omitted due to space or stylistic constraints, the full gels can be submitted as SI figures.”

Reply: We thank the reviewer for the suggestion. Molecular weight markers have been added to Fig. 3b and Fig. 3e (page 8) as follows:

Reversed Fig.3b and Fig.3e

However, molecular weight markers were not added to Fig.4 due to space constraints, the full gels of each *cat*-GFP-POI were shown in Supplementary Fig. 9b, Supplementary Fig. 9e, Supplementary Fig. 9h, and Supplementary Fig. 9k.

"I also spotted a couple of minor points:

** P4.L73 - 1,062,864 is $\sim 10^6$*

** A couple of places - facilely. While technically correct, this word is rarely used in British English. Consider replacing it with "easily", which is also more accessible.*

** P16.L274 - "Inspired by the reported streamlined synthesis method" feels like an inappropriate way to cite a publication that shares 3 authors with the current manuscript. I think that "Building on our reported streamlined synthesis method" may be more appropriate."*

Reply: We thank the reviewer for those considerate suggestions and apologize for the oversight.

* We have amended the calculation mistake on page 4.

* The word "facilely" on page 4 and page 17 has been replaced with "easily" as suggested.

* We have also rephrased the sentence "Inspired by the reported streamlined synthesis method" to "Building on our reported streamlined synthesis method" as suggested on page 16.

Reviewer #2

"In this manuscript, Zhang and coworkers describe strategies to form single domain GFP catenanes and their successful implementation. Key to the formation of GFP catenanes is the use of a rewiring process to reconnect GFP's secondary motifs. The formation and characterization of resulting protein constructs were verified by SDS-PAGE analysis and size exclusion chromatography. Overall, this is an interesting demonstration of single domain GFP catenanes and publication of this work in Nature Communications is recommended. The authors are advised to address the following issues before this paper is published."

Reply: We thank the reviewer for the kind comments.

“1. In Fig. 3, when the catenane formation provides multiple products, an estimation of product selectivity will be useful. The presence of fluorescent motif in these catenane constructs can potentially make this quantification easier, but other simpler assays are also available.”

Reply: We thank the reviewer for the suggestion. We determined the product selectivity by SDS-PAGE and gel densitometry and calculated the average yield from different independent experiments. We have added “yielding 83% protein catenane (46% for [2]cat-GFP and 37% for [3]cat-GFP) after Ni-NTA purification” on page 7 and added “showing the yields of in vitro synthesis (~46% [2]cat-GFP and ~37% [3]cat-GFP, determined by gel densitometry)” in the caption of Fig.3b to emphasis the efficiency of the in vitro synthesis. The yields have also been added to the caption of Fig. 6b to show the efficiency of the in vivo synthesis.

“2. Further exploring the applications of these intriguing protein catenanes would be of interests to the community.”

Reply: We thank the reviewer for the suggestion. What the reviewer has suggested is truly important. In this paper, we preliminarily explored the synthesis and functional benefits of protein catenanes as a proof of concept. The enhancement in thermal stability, thermal resilience, and mechanical stability may unlock the applications of topological proteins in biotechnology, pharmacology, and industry. Meanwhile, we are trying to promote the development of therapeutic proteins and industrial enzymes based on protein catenanes. To demonstrate this, we have added “Endeavors are being made to promote the development of therapeutic proteins and industrial enzymes based on protein catenanes.” on page 20.

Reviewer #3

“The work from Zhiyu Qu very nicely describes the approach towards the creation of relict free single domain catenanes. To exemplify the approach, they choose the green fluorescent protein (here superfolder variant sfGFP). The manuscript is well written and consistently tells the story of the catenanes creation as well as their advantages. While the authors eventually fail to fully produce the catenane in vivo with high yield they nonetheless provide a number of insightful aspects to this field. The work is in general experimentally sound and with sufficient experimental controls.”

Reply: We thank the reviewer for the kind comments.

“1) The following is a general writing almost philosophical aspect. While I appreciate the charm of speaking of “parallel fold universes” (“that the linear proteins family can be mapped catenane protein family”) with the single current paper (and also the small

body of previous work) I do not know if this wording is really justified. It suggests that each protein can be relatively straightforward converted into a catenane twin. Secondly, the wording in the abstract of “current protein universe” when speaking about the linear protein world suggests clear evidence that another fold universe with the same players was preceding or is about to come. From my perspective this is not backed by current knowledge.

Either the authors need to back up even their “questions” (l. 34 ff.) with ample literature reviewing from the field of protein / fold evolution or should rather cut this whole notion from the abstract and introduction and discuss it in the discussion section of the manuscript. Beyond that while “design principles” are mentioned (l. 66 ff.) those are very vague and general and do not constitute a claim to generalization. Also, such a claim would require additional fold classes to be tackled with this method. Thus, given that the example chosen by the authors uses a protein with known reconstitution (Cabantous, 2005) I would like the authors to reduce this notion of generalization.”

Reply: We thank the reviewer for the critical comment. First, “that the linear proteins family can be mapped catenane protein family” suggests that, theoretically, proteins with proper intrinsic entanglements can be converted into a catenane twin through rewiring, probably in a complicated way rather than relatively straightforward. However, its practical synthesis can be challenging. We are sorry for the confusion and have rephrased the sentence on page 9 into “that the linear FP family can be mapped into the catenane FP family.” to narrow the context appropriately, and added the word “theoretically” on page 19 to avoid ambiguity.

Second, there are only limited examples of natural protein catenanes like the capsid of bacteriophage HK97 (*Science* **2000**, 289, 2129-2133) and *Pyrobaculum aerophilum* citrate synthase (PaCS) (*J. Mol. Biol.* **2007**, 368 (5), 1332-1344). The examples of artificial protein catenanes were also limited (*J. Am. Chem. Soc.* **2016**, 138 (43), 14214-14217, *Protein Sci.* **2007**, 16 (7), 1249-1256, *Angew. Chem. Int. Ed.* **2016**, 55 (10), 3442-3446, *Angew. Chem. Int. Ed.* **2019**, 58 (32), 11097-11104, *Angew. Chem. Int. Ed.* **2020**, 59, 2-8, *J. Am. Chem. Soc.* **2021**, 143 (43), 18029-18040). Therefore, there is little discussion in literature on the fold universe of protein catenanes. Nevertheless, the evolution and classification of knotted and lasso proteins have been reviewed comprehensively (*Nat. Prod. Rep.* **2012**, 29, 996-1006, *Proc. Natl. Acad. Sci. U. S. A.* **2017**, 114 (13), 3415-3420, *Comput. Struct. Biotechnol. J.* **2015**, 13, 459-468, *Curr. Opin. Struct. Biol.* **2020**, 60, 131-141,), which have been cited in the paper as Ref. 15, Ref. 16, Ref. 17, and Ref. 20.

Inspired by the diversity of topological proteins, we put forward the concept of “parallel fold universes”. In our lab, we have successfully converted at least 5 types of different linear folds into catenane folds. These results will be reported in due course. While we are confident and excited about this idea, we agree with the reviewer that the current reported data is still insufficient to fully backup this notion. Hence, we have rephrased the sentence in the abstract and introduction and moved the majority of this discussion

in the “Folds of Topological Proteins” section.

“2) The comparison to wt sfGFP is lacking a bit. While I see that a good and direct comparison is the *l*-GFP the authors modeled after the sequence of their catenane versions I am not sure how well the catenanes fare in comparison to a “native” protein. The authors mention wt comparisons only once regarding the refolding (Fig. 5 and Suppl. Fig. 18). Especially with this data I am quite surprised on the described low refolding of their wt based on the superfolder, since literature (Andrews, *J Mol Biol* , 2009, *GdnHCl*, though, *Temp* needs to be looked up) indicates almost complete refolding for the folder optimized versions of avGFP. The limited recovery of around 10% the authors state appears very low. I would ask the author to revisit that data and also add the broad literature on refolding studies of FPs to their text to put the behaviors they study in context.”

Reply: We thank the reviewer for the comment. The *l*-GFP is largely the same as wt sfGFP-OPT. We tried to make an ideal control with identical amino acid composition to that of the *c*-GFP and [2]*cat*-GFP. Extra amino acids (including leading sequence MGSS and HisTag for purification) were added to the N- and C-termini of wt sfGFP-OPT. Therefore, *l*-GFP and wt sfGFP-OPT are expected to behave almost the same. We have also repeated the experiments many times and we confirm that *l*-GFP only has ~10% recovery of fluorescence after heat-induced refolding.

To check the refolding of sfGFP from chemically induced denaturation, we also looked into the literature suggested by the reviewer (*J. Mol. Biol.* **2009**, 392 (1), 218-227). As shown in Fig. R1a (i.e. Fig. 1 in literature), after equilibrating to 1.8 M Gdn-HCl for 96 h to refold, the refolded fraction was still less than 40 %, which is far from complete. To verify our finding, we looked into the literature on the refolding of sfGFP after thermal denaturation. A similar phenomenon has been reported by Prof. David Liu and coworkers (*J. Am. Chem. Soc.* **2007**, 129 (33), 10110). As shown in Fig. R1b (Figure S2 in literature), sfGFP can barely refold after being boiled at 100 °C for 1 minute and cooled down to and annealed at room temperature for 2 hours. Although they did not monitor the recovery percentage quantitatively as we did, the remained fluorescence in Fig. R1b is equally dim, just as our result shown in Fig. 5c. We appreciate the reviewer’s suggestion. To put the study in context, we have added these references on page 12 as suggested.

Fig. 1. The folding of sfGFP exhibits hysteresis with a long-lived trapped intermediate. The non-coincidence of the simulated folding equilibrium curves shows hysteresis. The red circles show the preparation of the native sample by equilibration at 1.8 M Gdn-HCl for 96 h. The trapped sample is initially unfolded to 6.5 M Gdn and then equilibrated to 1.8 M Gdn-HCl for 96 h. Both samples are placed under NMR conditions by $2\times$ spin columns into 0 M Gdn-HCl.

Figure S2. Thermally induced aggregation of stGFP under alternative conditions. The starting GFP (stGFP) was mixed to final concentrations of 2.0 (a), 0.5 (b), or 0.1 (c) mg/mL in a buffer containing 10 mM Tris pH 7.5, 5 mM NaCl, 5 mM dithiothreitol (DTT), and 10% glycerol. The left panel shows the samples before heating, under UV illumination. Samples were heated for 1 minute at 100°C, then cooled to room temperature for 2 hours and photographed again under UV illumination.

Fig. R1 a, Gdn-HCl-induced denaturation and renaturation in *J. Mol. Biol.* **2009**, 392 (1), 218-227. **b,** Thermal-induced denaturation and renaturation in *J. Am. Chem. Soc.* **2007**, 129 (33), 10110.

“3) The authors do not comment much on the chromophore maturation. All spectral characterizations are normalized to the peak. I would like to see raw absorbance spectra including the 280 nm protein peak of all described constructs in comparison to wildtype to judge the effectivity of chromophore maturation not only protein folding.”

Reply: We understand the reviewer’s concern. we added the spectral characterizations as the reviewer suggested. We diluted all samples to a similar molar concentration, that is 10 μ M (except for [3]cat-GFP to 5 μ M since one [3]cat-GFP molecule has two chromophores), determined by the 280 nm protein peak. The UV absorption spectra including the 280 nm protein peak of wt-GFP, [2]cat-GFP, and [3]cat-GFP were added to Supplementary Fig. 5a. To meet the optimal detection range of the instrument, all samples were further diluted tenfold before fluorescence spectra measurement. The unnormalized fluorescence spectra of wt-GFP, [2]cat-GFP, and [3]cat-GFP were added to Supplementary Fig. 5b~d. Similarly, the UV absorption spectra including the 280 nm protein peak and the unnormalized fluorescence spectra of cat-GFP-POIs were added to Supplementary Fig. 11. The UV absorption spectra including the 280 nm protein peak and the unnormalized fluorescence spectrum of c-GFP were added to Supplementary Fig. 12d and 12e. The UV absorption spectra including the 280 nm protein peak and the unnormalized fluorescence spectra of c-GFP-POIs and l-GFP-POIs were added to Supplementary Fig. 13e~h, 14e~h, 15e~h, 16e~h. The UV absorption spectra including the 280 nm protein peak and the unnormalized fluorescence spectra of *VidaL*-cat-GFP were added to Supplementary Fig. 24d and 24e. These results suggest that chromophore maturation has effectively occurred.

We also comment on the chromophore maturation by adding “The fluorescence spectra of fusion protein catenanes suggested their chromophore maturation have effectively occurred (Supplementary Fig. 11).” on page 10, “The fluorescence spectra of all protein

samples also confirmed the chromophore maturation has effectively occurred (Supplementary Fig. 12-16).” on page 11, and “while the fluorescence spectrum of *VidaL-cat*-GFP confirmed the maturation of its chromophore (Supplementary Fig. 24).” on page 17 and 18.

The chromophore maturation was also supported by the LC-MS spectra where all samples show a loss of 20 Da in the final molecular weight consistent with the loss of one molecule of water and two hydrogen atoms during chromophore maturation.

“4) The authors claim additive stability effects of catenane based fusions. Since chimeric proteins are not limited to catenanes but often challenging it would be interesting to compare at least one of their chimeric constructs against a linear fusion based on a wt GFP where loop insertions are equally possible to judge the net effect of this approach.”

Reply: We thank the reviewer for the intriguing suggestion. In fact, we have also considered the possibility of using a linear or cyclic fusion based on a *wt*-GFP with loop-inserted POI. It is certainly possible and interesting. However, after careful consideration, we thought that it might not be appropriate for this work. This design would be very sensitive to the linker design between the two GFP fragments and the central POI, which shall complicate the comparison and distract the readers from the theme of our paper on a single-domain protein catenane. Hence, we would like to restrict our discussion to the design and synthetic strategy of single-domain protein catenane and the corresponding topological effect. Meanwhile, we will construct the design as the reviewer suggested and the results will be reported in a separate paper.

“5) The enzymatic activity test has only been done on one protein of interest fusion (DHFR). Have the others not been tried (why?) or do they give unsatisfactory results.”

Reply: We thank the reviewer for pointing these out. Four POIs were examined in our work, namely, leptin, DHFR, monellin, and mCherry, to test the approach’s tolerance to various structural scaffolds. Among them, DHFR is an enzyme and we have tested its catalytic activity; mCherry is a fluorescent protein and we have determined its fluorescence emission spectra. Both remain functional. Leptin is a cytokine and monellin is a sweet protein. Their biological activity relies on their binding with cognate receptors to trigger the downstream biological pathways, which involve cell or animal-level experiments and are quite complicated. It requires much more expertise and specialized facility beyond our reach. Therefore, we have not proceeded with the characterization of *cat*-GFP-leptin and *cat*-GFP-monellin. Our results on DSC suggest that leptin and monellin remain folded in the fusion and exhibit a T_m at ~ 55 °C and ~ 70 °C (Supplementary Fig. 18). They probably keep the original fold and shall be equally active, just as DHFR and mCherry. Since the focus of this work is on the design and synthesis of single-domain protein catenane, we would like to limit our exploration of *cat*-GFP-POI with emphasis on the proof-of-concept experiments. More in-depth

characterizations will be carried out and reported in due course in the future to illustrate the topological effects.

“6) The last major point is more out of interest. Chimeras on FPs are widely applied in sensor engineering (e.g. GCaMPs) for this the fusion to the protein(s) of interest is done in beta sheet 7 near the chromophore. Would such also work with the catenanes, could it be tried and might it add additional advantages to creating photoactive chimeras of FPs?”

Reply: We thank the reviewer for this brilliant idea. We do appreciate it. We completely agree that it would be exciting to develop a photoactive catenane chimera of FP. As far as we know, the high responsiveness of FP-based sensors relies on the high sensitivity to external signals changing the chromophore environment (e.g. circularly permuted structure would impart greater mobility, *Nat. Biotechnol.* **2018**, 36 (8), 726-737, *Nat. Methods* **2020**, 17 (11), 1139-1146). While the POI and the FP in the catenane form would exhibit strong conformational coupling, we anticipate that the effect will be tricky and subtle. We will look into this possibility and try the idea on both catenane form and pseudorotaxane form. In this case, it will be a completely different story and is better to be reported separately.

“Minor points:

l. 78: what does “representative” 10 designs vs. “we designed many” mean”

Reply: We thank the reviewer for the concern and apologize for the confusion. We rephrased the sentence as “We came up with 10 possible catenane designs” on page 5.

“l. 98: what does “Among numerous trials,” mean. Pls. specify and also inform on unsuccessful trials and their properties”

Reply: We thank the reviewer for the suggestion. We have added representative synthetic results of other designs in Supplementary Fig. 2k. In brief, the synthetic efficiency of Design A, Design B, and Design C were preliminarily evaluated in vivo using the gene cassette IntC1-(GFP1-a/b/c)-IntN1-IntC2-(GFP2-a/b/c)-IntN2. Only samples of Design A showed obvious green fluorescence after Ni-NTA purification. SDS-PAGE analysis of the recombination efficiency of Design A, Design B, and Design C after Ni-NTA purification in Supplementary Fig. 2k shows that the target molecular weight of GFP catenane (~30k Da) could only be identified in the lane of Design A, indicating the unsatisfied recombination efficiency in Design B and Design C.

Supplementary Fig. 2k

“Supp Fig 18f: Difference between legend and figure (black and green)”

Reply: We thank the reviewer for the concern and apologize for the mistake. We have revised the legend.

“Most figures: Error bars are not commented on. Are those from independent productions of the protein and then measurement (recommended) or repeated measurements?”

Reply: We thank the reviewer for the concern. They are from independent productions of the protein and then measurement. We have commented on this in the Method part and after Fig. 5, Supplementary Fig. 19, Supplementary Fig. 20, and Supplementary Fig. 22.

We thank once again the reviewers for their very helpful suggestions and you for your time in handling our manuscript. We sincerely hope that these revisions are satisfactory and we look forward to hearing from you.

Many thanks again with our best regards,

Yours sincerely,

Wen-Bin Zhang

Reviewers' Comments:

Reviewer #1:

Remarks to the Author:

I thank the authors for their corrections. I have a few more minor comments but I leave at the discretion of the editor whether those require a revision, or can simply be addressed during typesetting.

P1.L12: "Macromolecular...chemistry." The sentence, as is, is confusing. Maybe: Macromolecular catenanes that cooperatively fold into a single domain do not belong to the current protein universe, and their design and synthesis open new territories in chemistry.

P4.L65: "... closely resemble those native ones..." maybe clearer as "... closely resemble the native ones..."

P9.L161: "...FP family in the parallel protein universe." maybe more accurate as "...FP family in a parallel protein universe."

Table 1: Data are currently presented without error. Please include an estimate of the error in the measurement.

Fig 5 – Errors are shown as standard deviation but based on the author's rebuttal, these are measurement between different protein batches, which would therefore allow standard error of the mean to be displayed.

Other than that, I have no reservations in recommending publication.

Reviewer #2:

Remarks to the Author:

In this revised manuscript, Zhang and coworkers addressed my questions. I support the publication of this work in Nature Communications.

Reviewer #3:

Remarks to the Author:

I thank the authors for addressing my concerns - very nice work.

Reviewer #1

“I thank the authors for their corrections. I have a few more minor comments but I leave at the discretion of the editor whether those require a revision, or can simply be addressed during typesetting.”

Reply: We thank the reviewer for the considerate suggestions.

“P1.L12: ‘Macromolecular...chemistry.’ The sentence, as is, is confusing. Maybe: Macromolecular catenanes that cooperatively fold into a single domain do not belong to the current protein universe, and their design and synthesis open new territories in chemistry.”

Reply: We thank the reviewer for the suggestion. We have revised the sentence “Macromolecular ... chemistry.” as suggested.

“P4.L65: ‘... closely resemble those native ones...’ maybe clearer as ‘... closely resemble the native ones...’”

Reply: We thank the reviewer for the suggestion. We have revised the sentence on P4 as suggested.

“P9.L161: ‘...FP family in the parallel protein universe.’ maybe more accurate as ‘...FP family in a parallel protein universe.’”

Reply: We thank the reviewer for the suggestion. We have revised the sentence on P9 as suggested.

“Table 1: Data are currently presented without error. Please include an estimate of the

error in the measurement.”

Reply: We thank the reviewer for the suggestion. Each sample was only tested once since DSC experiments using MicroCal PEAQ-DSC are with high reproducibility but time-consuming and expensive. Therefore, we estimated the error according to the instrument parameters on www.malvernpanalytical.com. To show the possible maximum error, we combined the measurement reproducibility (< 0.18 °C according to the official guide) and system reproducibility (< 0.1 °C according to the official guide) to obtain an estimated error < 0.21 °C (calculated according to the error formula $\sqrt{(0.18^2+0.1^2)}$). To clarify this, we added “According to the official guide of MicroCal PEAQ-DSC, the measurement reproducibility gives an error of < 0.18 °C and the system reproducibility gives an error of < 0.1 °C. Therefore, the total error of our experiment would be < 0.21 °C according to the error formula.” in the legend of Table 1 and Supplementary Table 2.

“Fig 5 – Errors are shown as standard deviation but based on the author’s rebuttal, these are measurement between different protein batches, which would therefore allow standard error of the mean to be displayed.”

Other than that, I have no reservations in recommending publication.”

Reply: We thank the reviewer for the suggestion. For consistency, we prefer to keep the standard deviation, which is also commonly used in the literature showing the fluorescence signal variation and enzyme activity (see, for example, Ref [1] and Ref [2] published in *Nat. Chem. Biol.* monitoring the fluorescence signal variation and Ref [3] and Ref [4] published in *Nature* and *Nat. Commun.* showing the enzyme activity of PETase).

Reviewer #2

“In this revised manuscript, Zhang and coworkers addressed my questions. I support the publication of this work in Nature Communications.”

Reply: We thank the reviewer for the kind comments.

Reviewer #3

“I thank the authors for addressing my concerns - very nice work.”

Reply: We thank the reviewer for the kind comments.

We thank once again the reviewers for their very helpful suggestions and you for your time in handling our manuscript. We sincerely hope that these revisions are satisfactory and we look forward to hearing from you.

Many thanks again with our best regards,

Yours sincerely,

Wen-Bin Zhang

Reference

1. Zhang, S. & Ai, H.W. A general strategy to red-shift green fluorescent protein-based biosensors. *Nat. Chem. Biol.* (2020).
2. Styles, M.J. et al. Autoinducer-fluorophore conjugates enable FRET in LuxR proteins in vitro and in cells. *Nat. Chem. Biol.* (2022).
3. Tournier, V. et al. An engineered PET depolymerase to break down and recycle plastic bottles. *Nature* **580**, 216-219 (2020).
4. Chen, Z. et al. Biodegradation of highly crystallized poly(ethylene terephthalate) through cell surface codisplay of bacterial PETase and hydrophobin. *Nat. Commun.* **13**, 7138 (2022).